# Phosphorus(III)-assisted regioselective C–H silylation of heteroarenes

Dingyi Wang[1,4], Xiangyang Chen [2,4], Jonathan J. Wong [2], Liqun Jin[3], Mingjie Li[1], Yue Zhao [1], K. N. Houk [2✉] & Zhuangzhi Shi [1✉]

Heteroarenes containing carbon–silicon (C–Si) bonds are important building blocks that play an important role in the construction of natural products, pharmaceuticals, and organic materials. In this context, the C–H silylation of heteroarenes is a topic of intense interest. Indole C–H silylation can preferentially occur at the nucleophilic C3 and C2 position (pyrrole core), while accessing the C4-C7 positions (benzene core) of the indole remains highly challenging. Here, we show a general strategy for the regioselective C7-H silylation of indole derivatives. Mainly, the regioselectivity is determined by strong coordination of the palladium catalyst with phosphorus (III) directing group. Using this expedient synthetic strategy, the diverse C7-silylated indoles are synthesized effectively which exhibits the broad functional group compatibility. Moreover, this protocol also been extended to other heteroarenes such as carbazoles. The obtained silylated indoles have been employed in various transformations to enable the corresponding differently functionalized indole derivatives. Significantly, a cyclopalladated intermediate is successfully synthesized to test the hypothesis about the P (III)-directed C–H metalation event. A series of mechanistic experiments and density functional theory (M06-2X) calculations has shown the preferred pathway of this directed C–H silylation process.

[1] State Key Laboratory of Coordination Chemistry, Chemistry and Biomedicine Innovation Center (ChemBIC), School of Chemistry and Chemical Engineering, Nanjing University, Nanjing 210093, China. [2] Department of Chemistry and Biochemistry, University of California, Los Angeles, Los Angeles, CA, USA. [3] College of Chemical Engineering, Zhejiang University of Technology, Hangzhou, China. [4] These authors contributed equally: Dingyi Wang, Xiangyang Chen. ✉email: houk@chem.ucla.edu; shiz@nju.edu.cn

S ilylation of the C–H bond[1-3] is one of the most notable advances in the C–H functionalization field[4-11] and it has been widely employed in the preparation of organosilicon compounds[12-20]. Heteroarylsilanes are considered as an important and versatile intermediates for the construction of complex molecules, since silyl groups can be easily transferred into a wide varieties of substituents[21-23]. Thus, C–H silylation of heteroarenes have extensively been employed for the preparation of highly important silylated heteroarenes[24-29]. Due to the presence of multiple C–H bonds in heteroaromatic compounds, the site-selective C–H bond functionalization is represents a key challenge in this arena. For example, the most familiar indole heteroarenes would eventually undergo electrophilic aromatic substitution (SEAr) at C3-position (Fig. 1a). As early in 1984, Simchen and coworkers developed a method on regioselective electrophilic C3-silylation of indoles using $Me_3SiOTf$[30]. Recently, several research groups have certainly reported C3-selective C–H silylation of indoles by merging cooperative Si-H bond activation and $S_EAr$ reaction using catalytic amount of cationic Ru(II) complex[31], Brønsted acid[32], and B$(C_6F_5)_3$[33]. Notably, some of the elegant protocols have also been involved for the site-selective C–H silylation of indoles at C2-position (Fig. 1b)[34]. In 2008, Falck and co-workers described an iridium-catalysed C–H silylation of *N*-unsubstituted indoles at the C2 position under mild condition using modest excess of $Et_3SiH$ reagent[35]. Similarly, in 2015, Stoltz and Grubbs reported a method to access C2-silylated indoles by $KO^tBu$-catalysed C–H silylation[36,37]. In this context, C–H silylation occurs preferentially at the nucleophilic C3 or C2 position, while accessing the benzene core of the indoles remains a great challenge[38-40].

Usually, C7-selective C–H functionalization of indoles demands the installation of a functional group at the C2 position to block this possible site[41]. Notably, numerous research groups have recently made valuable progress on direct C–H functionalization of indoles at C7-position[42-49]. In 2010, the C7-selective C–H borylation of indole was uncovered through iridium catalysis with the assistance of *N*-silyl-directing group[50]. Our group[51,52] and Ingleson group[53] have reported the chelation-assisted C–H borylation mediated by $BBr_3$, in which the

installation of a pivaloyl group at the N1 position of indole selectively brings the boron species to the C7 position and allows subsequent C–H borylation in an efficient manner (Fig. 1c). Unlike the well-developed C–H borylation reactions, C7-silylation can only resort to C7-lithiated indoles with silicon electrophiles, this process is not compatible with many sensitive functional groups[54-57]. Herein, we report a regioselective C–H silylation of indoles at C7 position using organosilane reagents enabled by $Pd(OAc)_2$ with oxidative conditions (Fig. 1d)[58-60]. The key to this high regioselectivity is the appropriate choice of N-P$^tBu_2$ as a directing group and using 2,5-dimethyl-1,4-benzoquinone (DMBQ) as an external oxidant. The oxidant DMBQ can not only regenerate the catalyst, but also suppress the oxidation of the phosphorus (III) directing group.

## Results

**Reaction design.** Our study commenced with the optimization of the reaction of indole **1a** and hexamethyldisilane (**I**) (Table 1). After extensive experimentation, using $Pd(OAc)_2$ (10 mol%) as the catalyst and DMBQ (3.5 equiv) as the oxidant showed the best reactivity within 72 h at 120 °C without any external ligands, providing the C7-silylation product **2a** in 80% yield, in which the C2 and C3 silylation isomers **3** and **4** were not observed (based on GC; entry 1). Moreover, treatment of DMBQ with indole **1a** did not result in conversion to the oxidized by-products **5** and **6**, showing that N-P$^tBu_2$ substituent was tolerated under the conditions. Indeed, the selection of a suitable oxidant was found to have a dramatic impact on the transformation. The reaction was conducted with 1,4-benzoquinone (BQ) as the oxidant, resulting in considerably lower yield of **2a** (entry 2). Treatment of indole **1a** either with both $Ag_2CO_3$ (entry 3) or $Cu(OAc)_2$ (entry 4) showed much less efficient for C7-silylation, along with large amount of the by-product **5**. Using other palladium sources such as $PdCl_2$ led to a slight decrease in yield (entry 5). Other transition metal catalysts like [Rh(cod)Cl]$_2$, [RuCl$_2$(p-cymene)]$_2$ or [Ir(COD)Cl]$_2$ were completely unsuccessful for this transformation, showing the uniqueness of palladium catalyst (entry 6-8). Changing the

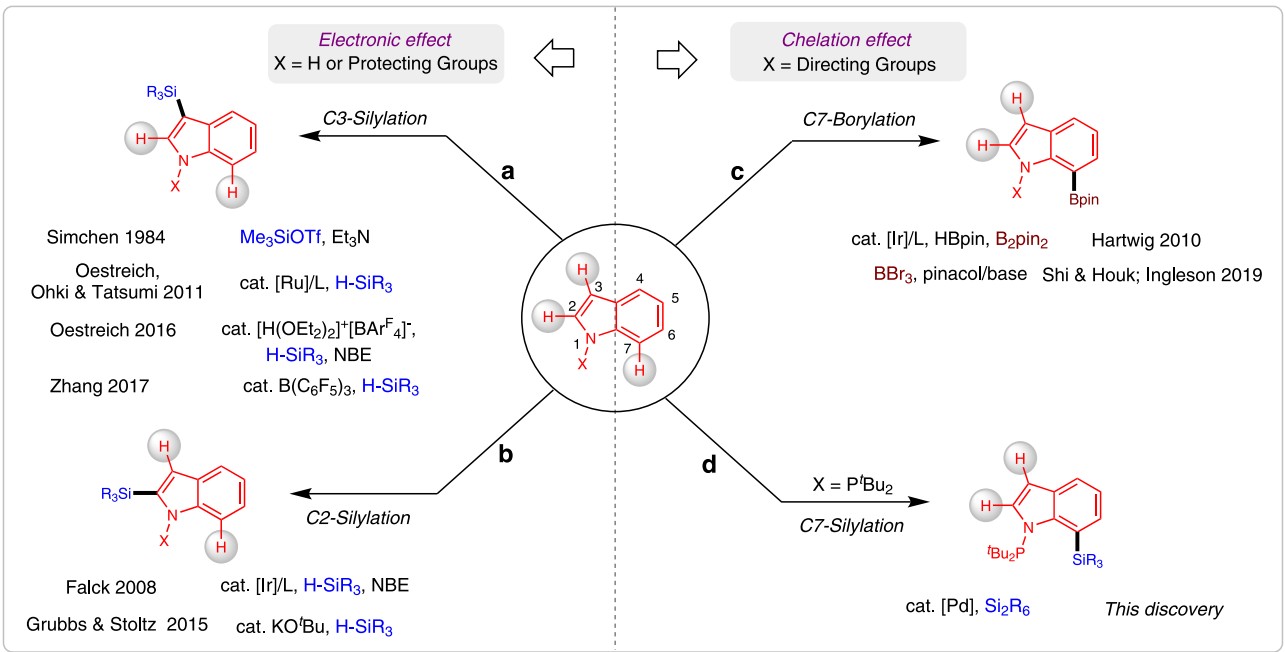

**Fig. 1 Development of a protocol to access C7-selective C–H silylation of indoles. a** C–H Silylation of indoles at C2-position. **b** C–H Silylation of indoles at C2-position. **c** C–H Borylation of indoles at C7-position. **d** C–H Silylation of indoles at C7-position.

**Table 1 Optimization of the reaction conditions[a].**

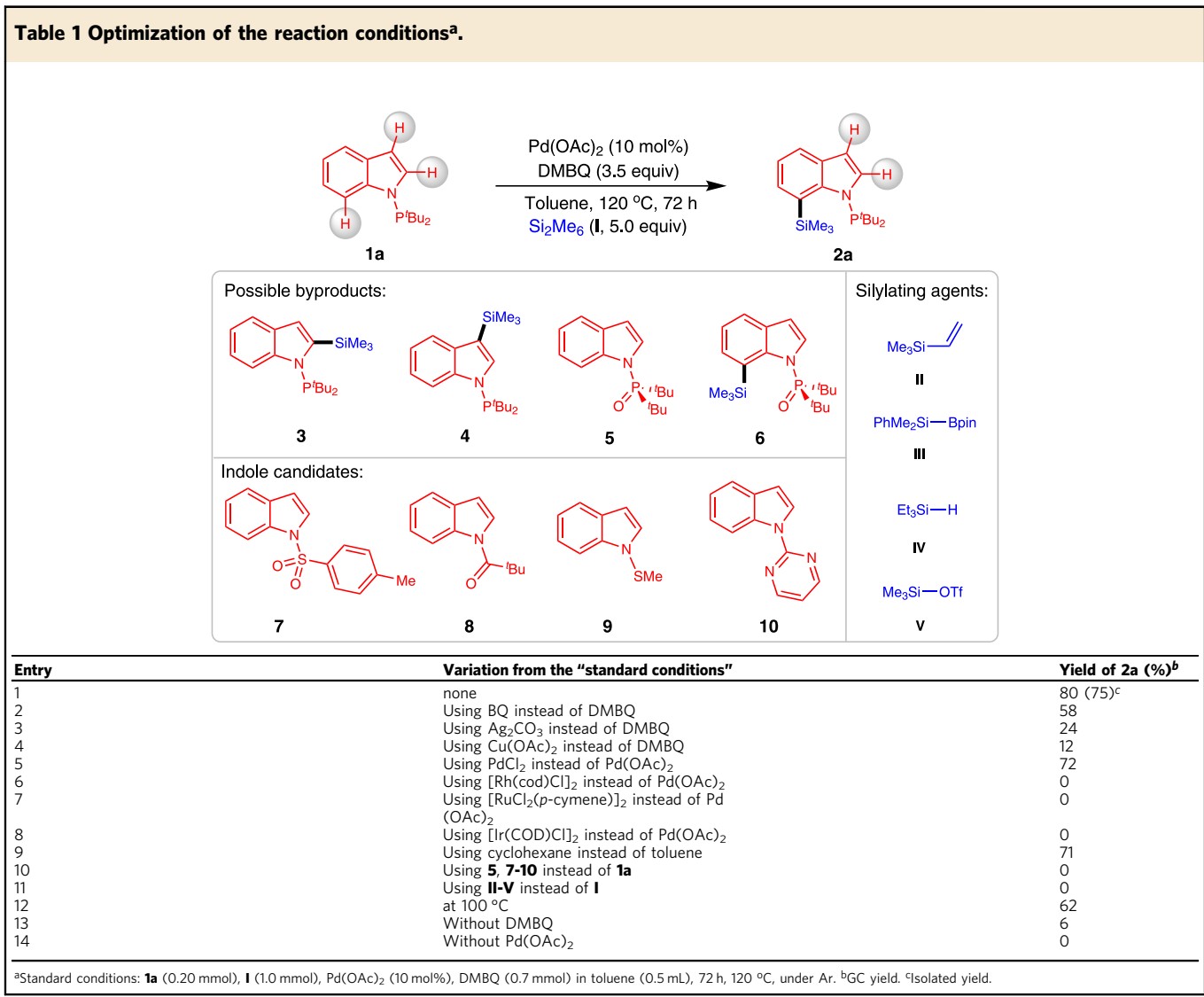

| Entry | Variation from the "standard conditions" | Yield of 2a (%)[b] |
|---|---|---|
| 1 | none | 80 (75)[c] |
| 2 | Using BQ instead of DMBQ | 58 |
| 3 | Using $Ag_2CO_3$ instead of DMBQ | 24 |
| 4 | Using $Cu(OAc)_2$ instead of DMBQ | 12 |
| 5 | Using $PdCl_2$ instead of $Pd(OAc)_2$ | 72 |
| 6 | Using [Rh(cod)Cl]$_2$ instead of $Pd(OAc)_2$ | 0 |
| 7 | Using [RuCl$_2$(p-cymene)]$_2$ instead of Pd(OAc)$_2$ | 0 |
| 8 | Using [Ir(COD)Cl]$_2$ instead of $Pd(OAc)_2$ | 0 |
| 9 | Using cyclohexane instead of toluene | 71 |
| 10 | Using **5**, **7-10** instead of **1a** | 0 |
| 11 | Using **II-V** instead of **I** | 0 |
| 12 | at 100 °C | 62 |
| 13 | Without DMBQ | 6 |
| 14 | Without $Pd(OAc)_2$ | 0 |

[a]Standard conditions: **1a** (0.20 mmol), **I** (1.0 mmol), $Pd(OAc)_2$ (10 mol%), DMBQ (0.7 mmol) in toluene (0.5 mL), 72 h, 120 °C, under Ar. [b]GC yield. [c]Isolated yield.

solvent to other nonpolar solvent like cyclohexane also provided product **2a** efficiently (entry 9), and the use of polar solvents involving DMF and THF, only led to trace amount of product **2a** (not shown in the table). The indole substrates bearing other directing groups like N-P(O)$^t$Bu$_2$ (**5**), N-Ts (**7**), N-Piv (**8**) and N-SMe (**9**)[49] were failed to generate any C–H silylation products, and only a small amount of C2 silylation by-product was obtained with 1-(pyrimidin-2-yl)-1H-indole (**10**) (entry 10). These results confirm the importance of the N-P$^t$Bu$_2$ group moiety for achieving both high reactivity and selectivity. Other silane reagents including trimethyl(vinyl)silane (**II**), SiMe$_2$Ph-Bpin (**III**), HSiEt$_3$ (**IV**), and TfOSiMe$_3$ (**V**) previously showed for the transition-metal-catalysed C–H silylations, but they completely failed in this reaction (entry 11). In addition, it was also found that the reaction temperature plays a substantial impact on this conversion (entry 12). As expected, the optimization studies clearly demonstrated that the reaction parameters including external oxidant DMBQ (entry 13) and palladium catalyst (entry 14) were optimal for the high conversion.

**Scope of the methodology**. We then investigated the scope of the palladium-catalysed C7-selective C–H silylation of indoles (Fig. 2). Indoles bearing methyl (**2b–d**) and phenyl (**2e–g**) substituents at

the C4-5 positions underwent facile C–H silylation and afforded the corresponding products in 45–77% yields. Among them, the structure of compound **2b** was confirmed by X-ray analysis (Supplementary Data 1). Indole substrates containing electron-donating groups such as OMe (**2 h** and **2i**), OBn (**2j**) were well tolerated, and a phenylthio group (**2k**) that remained without oxidation or deactivation of the catalyst. The synthetically useful halogens such as F (**2 l** and **2 m**) and Cl (**2n** and **2o**) were compatible. We further observed that C7 silylation of electronically deficient CF$_3$-substituted indole **2p** was also operative. Indole compounds bearing other electron-withdrawing groups like acetyl (**1q**), and ester (**1r** and **1 s**) formed the corresponding products **2q-s** in 52–80% yields. Noticeably, the strong coordination ability of cyano group (**1t-v**) with metal catalyst didn't inhibit the reaction outcome. In addition, indolylsilanes containing alkenyl (**2w**) or alkynyl (**2x**) substituent were also readily prepared using this methodology.

**Synthetic Applications**. To showcase the practical utility of this C–H silylation process, further investigations were conducted (Fig. 3). First, the functionalization of complex substrates was viable for phosphorus(III)-assisted regioselective C–H silylation reaction (Fig. 3a). In order to demonstrate the potential of this

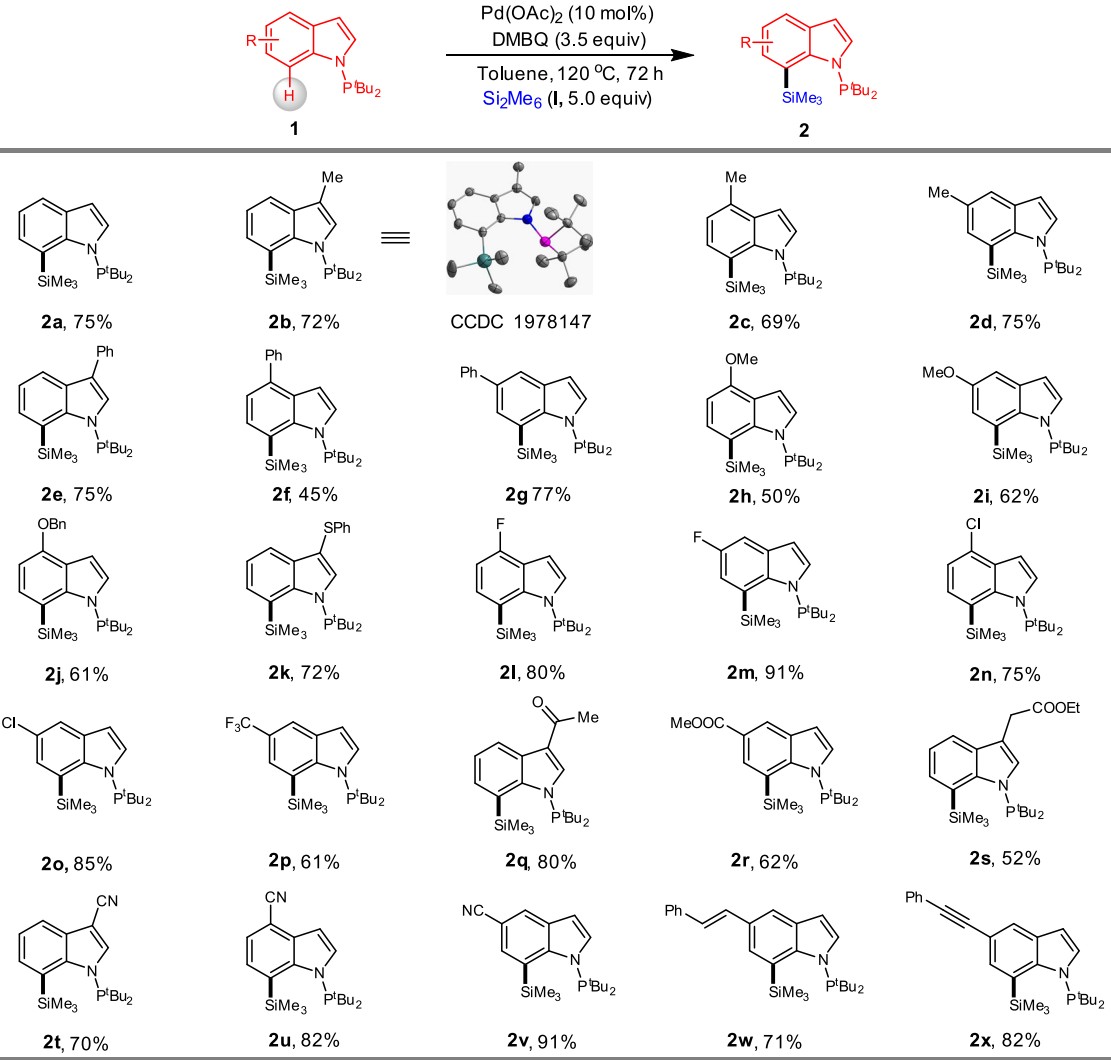

**Fig. 2 Substrate Scope.** Reaction conditions: **1** (0.20 mmol), **I** (1.0 mmol), Pd(OAc)$_2$ (10 mol%), DMBQ (0.7 mmol) in toluene (0.5 mL), 72 h, 120 °C, under Ar. All reported yields are isolated yields.

transformation, the complex indole molecules **11a–d** were subjected to the C7-silylation protocol. The reactions proceeded smoothly, yielding the Si-containing desired compounds **12a–d** in 61–76% yield with excellent chemo- and regioselectivity. The ketone, ester, and olefin moieties were well tolerated without any observed side reactions. Silylation of complex molecule **11d** with a Michael accepter motif also proceeded well without affecting the sensitive dienone motif. Second, the silylation reaction can be scalable to conduct and they are suitable to undergo various subsequent transformations, which shows the great potential in synthetic utility (Fig. 3b). A gram-scale reaction was conducted with indole **1a**, and the C7-silylated product **2a** was obtained in 66%. This compound could selectively get oxidized to phenol **13** and indolylsilane **4** bearing the $N$-P(O)$^t$Bu$_2$ unit in the presence of H$_2$O$_2$ in excellent yields, and the former one was further confirmed by X-ray analysis (Supplementary Data 1). With the compound **4** in hand, a variety of powerful synthetic transformations are also demonstrated. For instance, the Suzuki–Miyaura cross-coupling of **4** with different heteroaryl bromides **14a–c** by Si-B exchange furnishes 7-heteroarylated indoles **15a–c** in 45–65% yields[61]. Further deprotection of thiophen-indole **15c** with TBAF under mild condition could form N-H free indole **16** in a 70% yield. The effective method to form boronate ester **17**

was also demonstrated to proceed in good yield from silylated precursor **4**. In addition, the $N$-P(O)$^t$Bu$_2$ group in indole **4** could be removed to afford N-H free indole **18** in a 72% yield, which further undergo C2 selective C–H silylation by iridium catalyst, generating a bis-silylation product **19** in good yield. Third, this strategy can also be applied to site-selective C–H silylation of carbazoles (Fig. 3c). Considering the importance of carbazole and its derivatives widely used in functional materials, we further converted carbazoles **20a–d** into corresponding silylated products in 41–73% yields by palladium catalysis. Finally, the extension of the present protocol was further carried out on C7 selective C–H germanylation of indoles. Without changing the optimized condition, the reaction was conducted by employing indole **1a** with hexamethyldigermane (**VI**) gave the desired product **22** in a moderated yield (Fig. 3d).

## Discussion

To establish the mechanism for this C–H silylation process, we performed several mechanistic experiments (Fig. 4). When 1.0 equiv of indole **1a** was allowed to react with stoichiometric Pd (OAc)$_2$ in toluene at room temperature for 2 h in the absence of Si$_2$Me$_6$ (**I**), a cyclometalated Pd$^{II}$ dimer **23** was obtained and

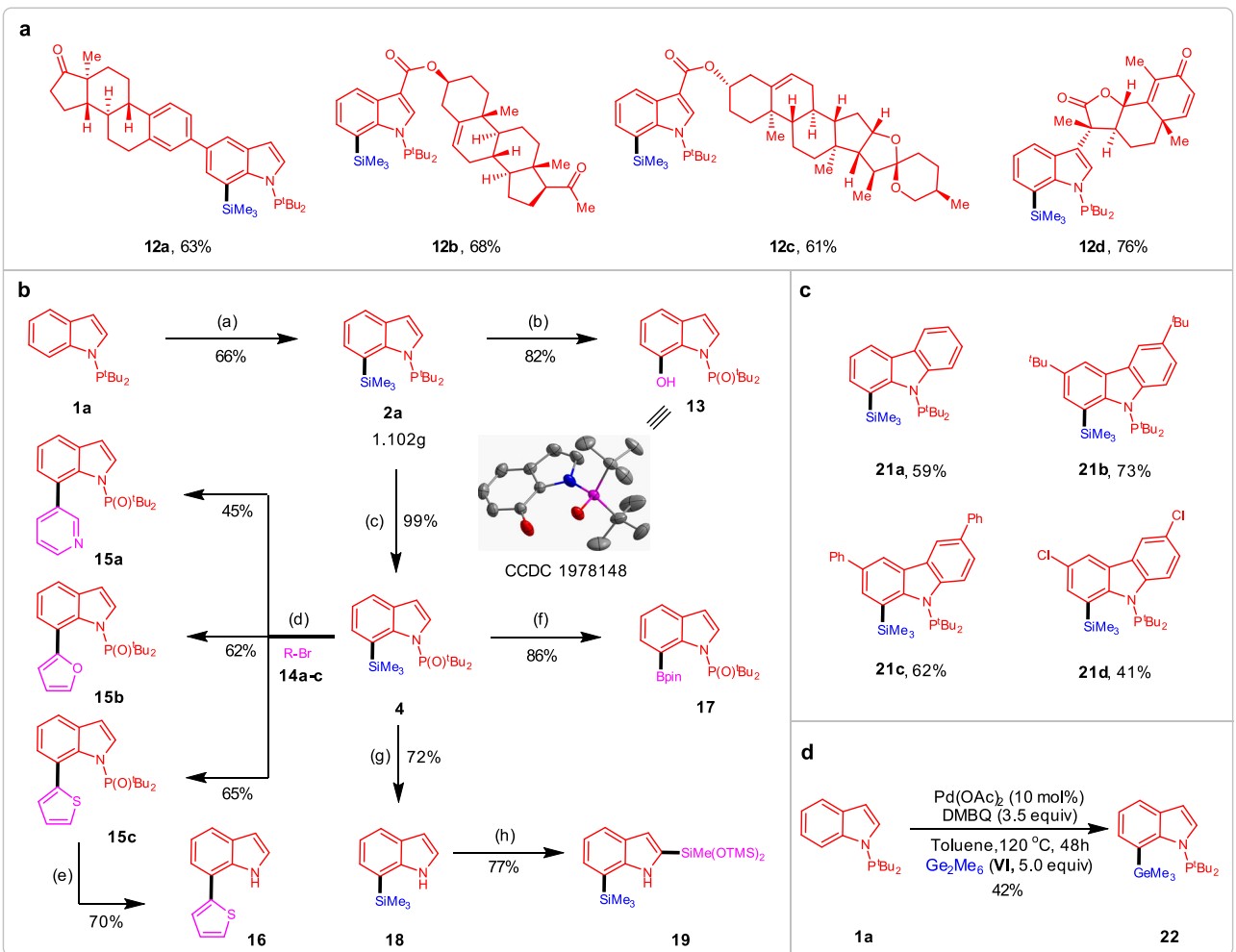

**Fig. 3 Further investigations. a** C–H Functionalization of complex molecules. **b** Downstream transformations. **c** C–H Silylation of carbazoles. **d** C–H Germylation of indole **1a**. Reagents and conditions: (a) **1a** (5.0 mmol, 1.0 equiv.), **I** (30 mmol, 6.0 equiv.), Pd(OAc)$_2$ (10 mol%), DMBQ (20 mmol, 4.0 equiv.) in toluene (10 mL), 4.5 days, 120 °C, under Ar; (**b**) **2a** (0.20 mmol), H$_2$O$_2$ (2.0 mL), KF (0.8 mmol), KHCO$_3$ (0.8 mmol) in THF (1.0 mL), 72 h, 40 °C; **c** **2a** (0.20 mmol), H$_2$O$_2$ (1.0 mL) in THF (0.5 mL), 72 h, rt; **d** **2a** (0.20 mmol), BCl$_3$ (0.24 mmol, 1 M in DCM) in DCM (1.0 mL), 6 h, rt, under Ar; then solvent was removed and RBr **14** (0.24 mmol), Pd(PPh$_3$)$_4$ (5 mol%), DME (2 mL) and 2 M Na$_2$CO$_3$ aqueous solution (0.5 mL), 24 h, reflux, under Ar. **e** **15c** (0.2 mmol), TBAF (1 M in THF, 0.4 mL) in THF (2.0 mL), 100 °C, 24 h. **f** **4** (0.20 mmol), BBr$_3$ (0.24 mmol, 1 M in DCM) in DCM (1.0 mL), 2 h, 0 °C to rt, under Ar; then solvent was removed and pinacol (0.4 mmol) and NEt$_3$ (1.0 mmol) in DCM (1.0 mL) were added, 1 h, rt; **g** **4** (0.20 mmol), iPr$_2$AlH (8.0 equiv.) in THF (1.0 mL), 12 h, 100 °C, under Ar; **h** **18** (0.20 mmol), [Ir(OMe)(cod)]$_2$ (5 mol% mmol), dtbpy (10 mol%), HSiMe(OTMS)$_2$ (0.60 mmol), NBE (0.6 mmol) in THF (1.0 mL), 24 h, 80 °C, under Ar.

confirmed by X-ray analysis (Supplementary Data 1) (Fig. 4a). This complex displays square planar geometry stabilized by bridging acetate and N-P$^t$Bu$_2$ ligands. Furthermore, product **2a** could be generated by using catalytic amount of complex **23**. To gain deeper insight into the mechanism of the C–H silylation reaction, we performed kinetic investigation by using P NMR spectroscopy (Fig. 4b). Varying the concentration of Pd in the range of 0.4–40 mM, initial rates for the Pd(OAc)$_2$-catalysed C–H silylation were obtained, which suggested a half-order dependence on [Pd]. By analogy to Sanford's observation[62–64], we proposed the resting state of the catalyst was the dimer **23**, which entered the catalytic cycle by dissociation to a monomer. Moreover, the reaction showed a zero-order dependence both on [indole **1a**] and [DMBQ] and a first order dependence on [Si$_2$Me$_6$ (**I**)], indicating that the coordination of N-P$^t$Bu$_2$ to Pd was very fast, DMBQ was not involved in the catalytic cycle, and the oxidative addition of Si–Si bond to catalyst center was the likely rate-limiting step. Next, we performed the intermolecular kinetic

isotope effect (KIE) experiments using **1a** and **D-1a**, in which a small KIE value of 1.07 was observed (Fig. 4c)[65]. This observation further indicated that the C–H bond cleavage was not the rate-determining step of the reaction. Finally, we also sought to probe the role of DMBQ in this reaction. DMBQ-induced catalyst regeneration led to reduce DMBQ to the corresponding silylated hydroquinones, since compounds **24** and **25** could be observed in a mixture (Fig. 4d).

Based on the above mechanistic experiments, density functional theory (DFT) calculations were then conducted on the model reaction of indole **1a** with Si$_2$Me$_6$ (**I**) to better understand the mechanism of this C–H silylation process and the C7 selectivity in indoles (Supplementary Data 2) (Fig. 5)[66,67]. Coordination of the Pd catalyst with **1a** to create **INT1** is endergonic by 0.7 kcal/mol due to an unfavorable decrease in entropy. **INT1** then deprotonates the C7 position of **1a** through a concerted metalation-deprotonation (CMD) transition state involving the carboxylate, **TSI_A** with an activation barrier of

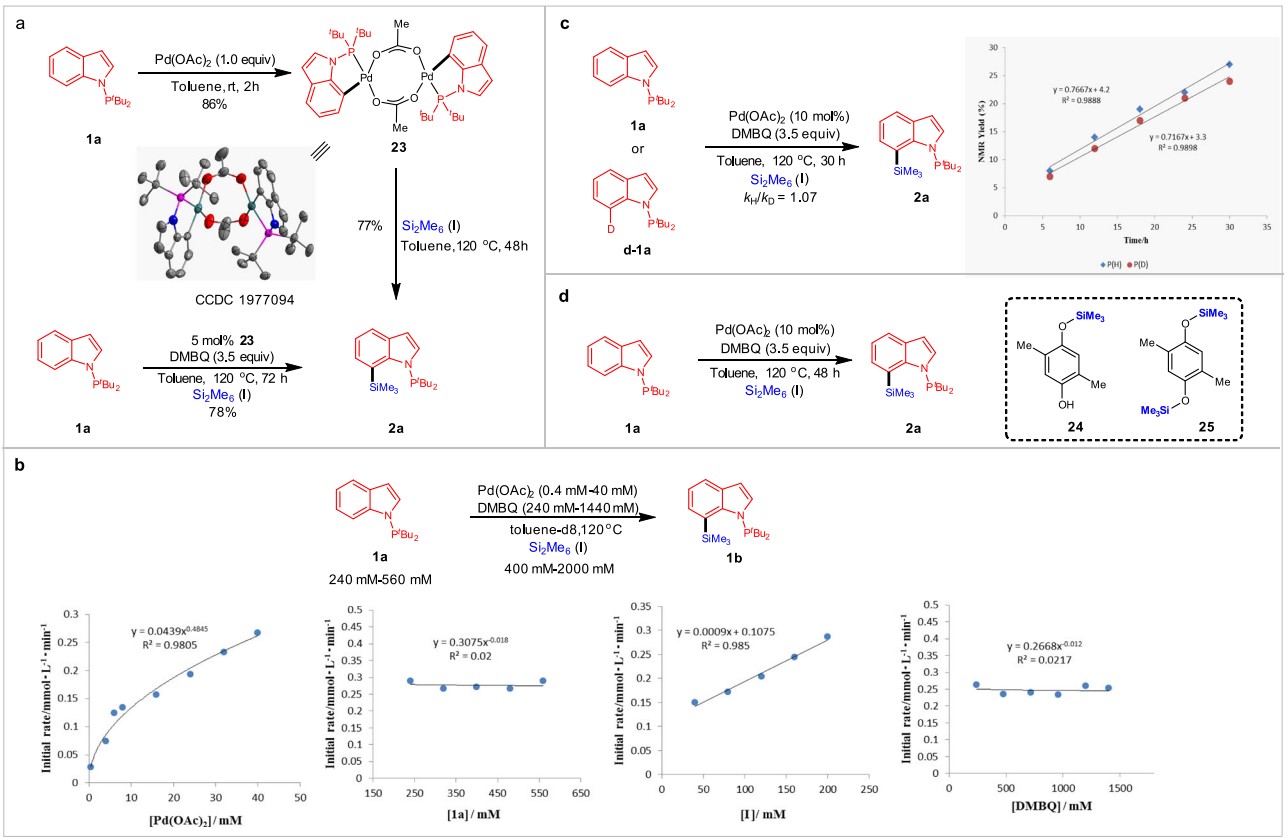

**Fig. 4 Mechanistic experiments. a** Cyclopalladation of indole **1a** to a bimetallic Pd(II) complex **23**. **b** Kinetic profiles of Pd-catalysed C−Hsilylation between substrates **1a** and **I**. **c** KIE experiments of **1a** and **d-1a**. **d** Investigation of the byproducts from DMBQ.

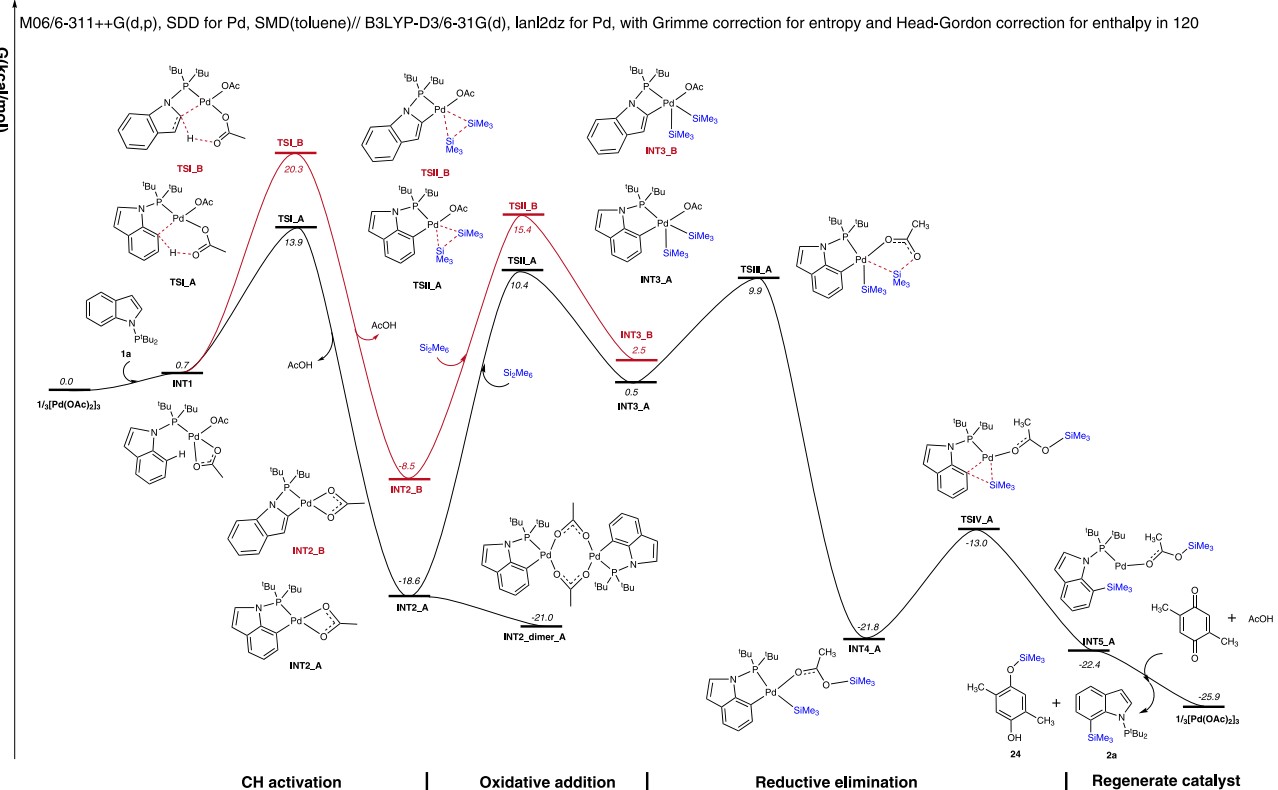

**Fig. 5 Free energy profiles for C2 (red) or C7 (black) selective C–H silylation of indole 1a.** DFT method: M06/6-311 ++G(d,p), SDD for Pd, SMD (toluene)// B3LYP-D3/6-31 G(d), lanl2dz for Pd, with Grimme correction for entropy and Head-Gordon correction for enthalpy in 120°C. All energies are in kcal/mol.

13.9 kcal/mol[68–71]. This leads to a stable intermediate **INT2_A** with a relative free energy of −18.6 kcal/mol. The intermediate **INT2_A** can further dimerize to the resting state **INT2_dimer_A**, which can be isolated from the system. The subsequent oxidative addition step inserts Pd into the Si-Si bond of reagent **I** and leads to the five-coordinated **INT3_A** through **TSII_A** with a free energy of 10.4 kcal/mol. The SiMe$_3$ then transfers the acetoxy group from Pd to the O of acetoxy through **TSIII_A** with a free energy of 9.9 kcal/mol to form the four-coordinated **INT4_A** (Fig. S1). The subsequent reductive elimination with the formation of C–Si bond proceeds easily with an active free energy of 8.8 kcal/mol with respect to **INT4_A**. According to the calculated free energy profile, the oxidative addition is the rate-determining step with a total energy barrier of 31.4 kcal/mol (**INT2_dimer_A → TSII_A**). The deprotonation from the C2 position was also studied. This leads to, however, a highly strained and unfavourable four-membered cyclic transition state **TSI_B** with a much higher free energy of 20.3 kcal/mol compared to its five-membered counterpart. This agrees with the experimental absence of the C2 silylation product **3**. A final dissociation would deliver the final product **2** while regenerating the active propagating Pd$^{II}$ species in the presence of DMBQ and in-situ formed AcOH.

In summary, we have explored a reliable strategy on regioselective C–H silylation of heteroarenes enabled by N-P$^t$Bu$_2$ directing group. This strategy shows numerous advantages including the using of commercially available catalyst, avoiding the addition of the exogenous ligand, and compatibility with broad substrate scope. Because of the ubiquity of the indole and carbazole frameworks and their precursors in biologically active compounds, we hope that this strategy based on the diversity of silicon chemistry will simplify the synthesis and structural elaboration of heteroarylsilanes for advanced research in chemistry, biology, and medicine.

## Methods

**General procedures for synthesis of 2**. In an oven-dried Schlenk tube, **1** (1.0 equiv, 0.20 mmol), Hexamethyldisilane **I** (5.0 equiv, 1.0 mmol), Pd(OAc)$_2$ (10 mol%, 4.48 mg, 0.02 mmol), DMBQ (3.5 equiv, 95.2 mg, 0.70 mmol) were dissolved in toluene (0.5 mL). The mixture was stirred at 120 °C under argon for 72 h. Upon the completion of the reaction, the solvent was removed. The crude mixture was directly subjected to column Chromatography on silica gel using petroleum ether/ EtOAc as eluent to give the desired products **2**.

## Data availability

The authors declare that the data supporting the findings of this study are available within the article and its Supplementary Information Files as well as from the corresponding authors upon reasonable request. The crystallography data have been deposited at the Cambridge Crystallographic Data Center (CCDC) under accession number CCDC: 1978147 (**2b**), CCDC 1978148 (**13**), CCDC 1977094 (**23**), and can be obtained free of charge from www.ccdc.cam.ac.uk/getstructures.

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

## Acknowledgements
We thank the "1000-Youth Talents Plan", the National Natural Science Foundation of China (Grants 21672097) and the "Innovation & Entrepreneurship Talents Plan" of Jiangsu Province for their financial support.

## Author contributions
Z.S. conceived and designed the study, and wrote the paper. D.W., M.L., and M.W. performed the experiments. K.N.H. supervised the mechanistic study. X.C. and J.W. performed the DFT calculations. Y.Z. performed the crystallographic studies. L.J. assisted the kinetic experiments.

## Competing interests
The authors declare no competing interests.
