## [Peer Review File · Nature Communications]

REVIEWER COMMENTS

Reviewer #1 (Remarks to the Author):

Shi, Houk and co-workers describe a novel C7-H silylation of indoles catalyzed by Pd. The reaction is a new contribution in this field and might be significant enough for Nature Commun. A reasonable scope is demonstrated, as well as the conversion of the C-Si bond to other valuable functionality. The

However, the manuscript and SI have major shortcomings. This includes aspects of the scope, references and the rigor with which the SI is written. There are also problems with the writing itself, especially the frequent use of confusing or inaccurate terminology.

SCIENTIFIC POINTS

Table 1 shows "variation from standard conditions". There is no set of conditions labelled or described as "standard". This would help. Presumably, the authors mean the conditions under the reaction arrow at the top of Table 1. But this is not stated. There, the equivalents of DMBQ are mentioned, but not for Si₂Me₃, which is inconsistent.

The significance of the variations of conditions in Table 1 are not mentioned. What is the significance of testing cyclohexane as solvent? No comment is made about this.

Line 78: "other transition metal catalysts like Rh salts..."

This suggests various salts/catalysts, but apart from PdCl₂ only one variation is listed (entry 6). Were other catalysts tested? The authors do not say. This makes it hard to know what was done and how unique the Pd is. Were Ru or Ir salts tested? If not, why not? These are amongst the most widely used in C-H silylation to date. The authors should explain this.

Lines 87-89: "As expected, rigorous control experiments demonstrate that the reaction parameters including external oxidant DMBQ (entry 11) and palladium catalyst (entry 12) were crucial for the silylation to occur."

This is obviously inaccurate, since the authors show that the products could be obtained without DMBQ (e.g. entry 2 or 3), so it is not "crucial". It's just optimal.

Were C2 or C3 silylated products ever observed with the other directing groups? If the more weakly coordinating DGs were not good at directing the Pd to C7, maybe C3-palladation occurred? Maybe it occurred and no silylation occurred? Were any experiments performed to test this?

Have the authors considered placing a -P(tBu)₂ at C3, to effect C4-H silylation? If this did not work, the authors should say.

Scope: In general, the scope in Figure 2 is good. Many important functional groups are represented. However, despite what the authors write, the "practical utility" (line 104) is not demonstrated by products 12a-d. What is the practical value of these compounds? The authors do not explain. They are all based on a steroid unit, and all of these have few interesting functional group not represented in Fig 2, so I can't help but conclude that the "practical utility" is hype. However, the reaction scope in Fig 3B is very good. It is important to show that the C-Si bond can be converted to other types of bonds.

Line 116: The authors discuss the conversion of compound 4 to products 15 through a "Suzuki-Miyaura cross-coupling". But Suzuki reactions are with boronates, not silanes. Presumably the authors mean Hiyama coupling (although other named variations exist). This should be corrected.

The authors might also want to consider referencing work done on coupling C-Si reagents in this way, including work on coupling specifically trialkylsilyl arenes.

The significance of the mechanistic work, e.g. which step is rate-determining, would be useful to set in context. How do the mechanistic findings compare to other (hetero)arene C-H silylation methods?

The calculations indicate what looks to me like a CMD mechanism (e.g. TSI_A), even if it's not the rate-determining step. No reference is made to previous work on CMD, e.g. by Fagnou, Davies and others. If the acetate ligand is involved in the mechanism for C-H activation, how does the reaction work when PdCl₂ is used as the catalyst (table 1, entry 5)?

POINTS ABOUT WRITING

Quite a few references are missing.

For example:

J. Am. Chem. Soc. 2019, 141, 7063-7072.

Tetrahedron 2019, 75, 4059-4070.

Chem Commun. 2016, 52, 5868-5871

J. Am. Chem. Soc. 2016, 138, 25, 7982-7991

J. Org. Chem. 2019, 84, 5863-5871

Org. Biomol. Chem., 2017, 15, 4783

Angew. Chem. Int. Ed., 2017, 56, 13872

... and the papers by the other developers of other C-H silylation reactions, including Suginome, Kakiuchi and Chatani.

Line 11: "building blocks with complementary reactivity..."

Complementary to what?

Line 14: "at the electrophilic C3-C2 positions (pyrrole core),"

The authors repeat this on line 41: "electrophilic C3 or C2 position"

The C2 and C3 positions are nucleophilic. The authors even allude to this themselves on line 32 when they mention that these positions could be derivatized using SEAr. In SEAr, the arene is the nucleophile, not the electrophile.

Also what does "C3-C2" mean, as a form of nomenclature? (Line 14)

Lines 20-21: "A cyclopalladated intermediate was first confirmed by X-ray analysis."

What does "first" mean? This is ambiguous. Was this the first thing done? Is it the first such complex? Moreover, the sentence could be taken to mean that the intermediacy of the complex was confirmed by X-ray analysis, which presumably the authors don't mean.

Line 26: "Has been widely employed"

Where? When? By whom? No references are offered to back this up, either.

Line 54: "indoly lithium": this must be a typo.

Line 62: "N-PtBu₂ indole" is not correct nomenclature.

Lines 63-64: "To optimize the reaction conditions during experimentation, the best results were obtained..."

This is very unclear. Taken as written, it means the best results were obtained in order to optimize the reaction. Were *only* the best results obtained for this purpose? What are reaction conditions during experimentation? Can you have experimentation without reaction conditions. The

manuscript has many such confusing turns of phrase. A native English speaker should proof-read this manuscript and clarify all these points.

Here's another example:

Lines 67-68: The isomers involving C2 and C3 silylation products..."
Isomers don't "involve" products.

Line 68: "tolerable" should be "tolerated"

Line 73: "erosion" should really be "reduction"

The abbreviation DMBQ is never explained, which will disadvantage readers from outside of this field.

Line 82: "was" should be "were"

Line 83: "It" should be "This"

Line 106: "viable with late stage modification":

"late stage functionalization" (LSF) is more commonly used. Also the authors could cite some recent reviews here relating to LSF, e.g. Cernak's Chem. Soc. Rev., 2016, 45, 546-576, although there are several others.

Also, the phrase "[the] reaction was viable with late-stage modification" doesn't really mean anything. Presumably, the late-stage functionalization was viable.

Line 110: "products can be scalable to generate" is muddled. The reaction is scalable, not the products.

Line 116: After "demonstrated" please refer the reader to the appropriate scheme.

Line 141: N-PtBu should presumably have two tert-butyl groups.

Line 162: Should "Association" be "Coordination"?

Line 170: "the Si-Si bond of 2a". In Figure 4a, 2a is shown and has no Si-Si bond.

Line 184: "compatible" should be "compatibility" and "a broad range of substrate scope" is redundant; "broad substrate scope" is meant.

SUPPORTING INFO

There are problems in the rigor with which the SI is prepared. For example, in the section describing the synthesis of complex 23, the complex is labelled as 6, which is not consistent with the manuscript.

The description of this procedure for 23 (or "6") on page S27 says "The single crystals of 6 were grown by slow evaporation of the DCM/ Hexane." What DCM? None is mentioned.

For clarity, the spectra in the SI should be labelled. For example, for on page S79, I'm guessing that's 31P NMR and the spectrum on S80 is 19F NMR.

The 1H NMR spectrum of 2u on page S86 shows significant impurities, which presumably puts the yield of 2u (86%) in question. Also, what I presume is the 31P NMR of 2u (p S87) is very weak, and I note that the solvent peaks in the 1H and 13C NMR spectra of that compound are proportionally higher than for some of the other compounds. Similarly, the 1H NMR spectrum of 16 is not clean enough, so the yield for that is also presumably not the 74% reported.

Why is the spectrum at the top of page S98 zoomed into such a narrow ppm range? The whole spectrum should be showed - and the nucleus indicated - and an expansion provided if desired.

I stress that these are all *examples* of problems in the SI. Obviously, it's not possible to list all the others here. But those I listed above are representative and were quick to find, so the entire SI should be carefully revised and checked for rigor in every detail. If yields are not accurate, they should be recorded correctly.

Reviewer #2 (Remarks to the Author):

This manuscript presents the work on phosphorus(III)-assisted regioselective C-H silylation of heteroarenes studied experimentally and computationally, which includes that a cyclopalladated intermediate is first confirmed by X-ray analysis to support P(III)-assisted C-H silylation mechanism. The work seems publishable, since this is one type of important reactions. However, authors should consider the following concerns:

(1) For the DFT calculation, authors should address why zero yield is observed for reactants 7-10 in entry 8 of Table 1. At the same time, it will be much better to address the zero yield in entry 9 of Table 1.

(2) Authors should also complete the DMBQ-involved process, if too much for text, it could be included in SI, which might help to understand the data of entry 2-4 in Table 1.

(3) Authors should compare B3LYP results with M06.

Reviewer #1 (Remarks to the Author)

Shi, Houk and co-workers describe a novel C7-H silylation of indoles catalyzed by Pd. The reaction is a new contribution in this field and might be significant enough for Nature Commun. A reasonable scope is demonstrated, as well as the conversion of the C-Si bond to other valuable functionality. However, the manuscript and SI have major shortcomings. This includes aspects of the scope, references and the rigor with which the SI is written. There are also problems with the writing itself, especially the frequent use of confusing or inaccurate terminology.

Response: Thank you for the constructive comments, which are very helpful for us to improve the quality of this article.

SCIENTIFIC POINTS

Table 1 shows “variation from standard conditions”. There is no set of conditions labelled or described as “standard”. This would help. Presumably, the authors mean the conditions under the reaction arrow at the top of Table 1. But this is not stated. There, the equivalents of DMBQ are mentioned, but not for Si_2Me_6 , which is inconsistent.

Response: The equivalents of Si_2Me_6 are mentioned under the reaction arrow. In addition, the standard conditions are stated in the text.

The significance of the variations of conditions in Table 1 are not mentioned. What is the significance of testing cyclohexane as solvent? No comment is made about this.

Response: Only the nonpolar solvent can provide product 2a efficiently, thus cyclohexane is mentioned in the table. The use of polar solvents involving DMF and THF, only led to trace amount the product 2a. These two results were mentioned in the text but not showed in the table.

Line 78: “other transition metal catalysts like Rh salts...”

This suggests various salts/catalysts, but apart from PdCl_2 only one variation is listed (entry 6). Were other catalysts tested? The authors do not say. This makes it hard to know what was done and how unique the Pd is. Were Ru or Ir salts tested? If not, why not? These are amongst the most widely used in C-H silylation to date. The authors should explain this.

Response: Yes. Other catalysts widely used in C-H silylation like $[\text{RuCl}_2(\text{p-cymene})]_2$ or $[\text{Ir}(\text{COD})\text{Cl}]_2$ were completely failed in this transformation, showing the unique of palladium catalyst. We added the information in Table and text during the revision.

Lines 87-89: “As expected, rigorous control experiments demonstrate that the reaction parameters including external oxidant DMBQ (entry 11) and palladium catalyst (entry 12) were crucial for the silylation to occur.” This is obviously inaccurate, since the authors show that the products could be obtained without DMBQ (e.g. entry 2 or 3), so it is not “crucial”. It’s just optimal.

Response: We changed the description in the text.

Were C2 or C3 silylated products ever observed with the other directing groups? If the more weakly coordinating DGs were not good at directing the Pd to C7, maybe C3-palladation occurred? Maybe it occurred and no silylation occurred? Were any experiments performed to test this?

Response: We didn’t observe any C-H silylation products when using indoles bearing N-P(O)^tBu₂ (7), N-Piv (8) and N-SMe (10) groups. While indoles bearing N-pyrimidin-2-yl (9) was selected as

a substrate, a small amount of C2 silylation by-product was obtained.

Have the authors considered placing a $-P(tBu)_2$ at C3, to effect C4-H silylation? If this did not work, the authors should say.

Response: Thanks for this kind suggestion! Your proposed transformation looks very probable. However, as a practical directing group, it should be easy to install and remove from the substrates. We didn't try such transformation, because it's not easy to install and remove the $-P(tBu)_2$ at C3 position of indoles at the current stage.

Scope: In general, the scope in Figure 2 is good. Many important functional groups are represented. However, despite what the authors write, the "practical utility" (line 104) is not demonstrated by products 12a-d. What is the practical value of these compounds? The authors do not explain. They are all based on a steroid unit, and all of these have few interesting functional group not represented in Fig 2, so I can't help but conclude that the "practical utility" is hype. However, the reaction scope in Fig 3B is very good. It is important to show that the C-Si bond can be converted to other types of bonds.

Response: We changed the description in the text.

Line 116: The authors discuss the conversion of compound 4 to products 15 through a "Suzuki-Miyaura cross-coupling". But Suzuki reactions are with boronates, not silanes. Presumably the authors mean Hiyama coupling (although other named variations exist). This should be corrected. The authors might also want to consider referencing work done on coupling C-Si reagents in this way, including work on coupling specifically trialkylsilyl arenes.

Response: We changed the description in the text.

The significance of the mechanistic work, e.g. which step is rate-determining, would be useful to set in context. How do the mechanistic findings compare to other (hetero)arene C-H silylation methods?

Response: According to the calculated free energy profile, the oxidative addition is the rate-determining step with a total energy barrier of 31.4 kcal/mol (INT2_dimer_A \rightarrow TSII_A). In most reported works (Fig 1A & B), (hetero)arene C-H silylation proceeds via dehydrogenative process. In this work, the Pd-catalyzed C-H silylation of indoles at C7 position with organosilane reagents occurs through a concerted metalation-deprotonation (CMD) pathway under oxidative conditions.

The calculations indicate what looks to me like a CMD mechanism (e.g. TSI_A), even if it's not the rate-determining step. No reference is made to previous work on CMD, e.g. by Fagnou, Davies and others. If the acetate ligand is involved in the mechanism for C-H activation, how does the reaction work when PdCl₂ is used as the catalyst (table 1, entry 5)?

Response: The references on CMD mechanism have been added (refs 73-76). The mechanism using PdCl₂ as the catalyst is different with that of Pd(OAc)₂. In the absence of acetate, DMBQ can act as an external base to assist the deprotonation step. Compared to the CMD process, the external base way is impossible with a high free energy of 38.8 kcal/mol. But using the DMBQ as the external base in the presence of PdCl₂ for C-H activation is feasible with a lower free energy

barrier of 26.1 kcal/mol.

POINTS ABOUT WRITING

Quite a few references are missing.

For example:

J. Am. Chem. Soc. 2019, 141, 7063-7072. Tetrahedron 2019, 75, 4059-4070. Chem Commun. 2016, 52, 5868-5871. J. Am. Chem. Soc. 2016, 138, 25, 7982-7991. J. Org. Chem. 2019, 84, 5863-5871. Org. Biomol. Chem., 2017, 15, 4783. Angew. Chem. Int. Ed., 2017, 56, 13872.... and the papers by the other developers of other C-H silylation reactions, including Suginome, Kakiuchi and Chatani.

Response: The mentioned references and the papers by the other developers of C-H silylation reactions have been added in the manuscript (refs 4-12).

Line 11: "building blocks with complementary reactivity..." Complementary to what?

Response: We changed it.

Line 14: "at the electrophilic C3-C2 positions (pyrrole core),"

The authors repeat this on line 41: "electrophilic C3 or C2 position"

The C2 and C3 positions are nucleophilic. The authors even allude to this themselves on line 32 when they mention that these positions could be derivatized using SEAr. In SEAr, the arene is the nucleophile, not the electrophile.

Also what does "C3-C2" mean, as a form of nomenclature? (Line 14)

Response: We corrected them.

Lines 20-21: "A cyclopalladated intermediate was first confirmed by X-ray analysis."

What does "first" mean? This is ambiguous. Was this the first thing done? Is it the first such complex? Moreover, the sentence could be taken to mean that the intermediacy of the complex was confirmed by X-ray analysis, which presumably the authors don't mean.

Response: We changed the description.

Line 26: "Has been widely employed"

Where? When? By whom? No references are offered to back this up, either.

Response: We added the related references in the text.

Line 54: "indoly lithium": this must be a typo.

Response: We corrected it.

Line 62: "N-PtBu2 indole" is not correct nomenclature.

Response: We corrected it.

Lines 63-64: "To optimize the reaction conditions during experimentation, the best results were obtained..." This is very unclear. Taken as written, it means the best results were obtained in order to optimize the reaction. Were *only* the best results obtained for this purpose? What are reaction conditions during experimentation? Can you have experimentation without reaction conditions. The manuscript has many such confusing turns of phrase. A native English speaker should proof-read this manuscript and clarify all these points.

Response: Thanks for your suggestion! This manuscript has been modified by a native English speaker.

Here's another example:

Lines 67-68: The isomers involving C2 and C3 silylation products..."

Isomers don't "involve" products.

Response: We corrected it.

Line 68: "tolerable" should be "tolerated"

Response: We corrected it.

Line 73: "erosion" should really be "reduction"

Response: We corrected it.

The abbreviation DMBQ is never explained, which will disadvantage readers from outside of this field.

Response: We added the full name in the text.

Line 82: "was" should be "were"

Response: We corrected it.

Line 83: "It" should be "This"

Response: We corrected it.

Line 106: "viable with late stage modification": "late stage functionalization" (LSF) is more commonly used. Also the authors could cite some recent reviews here relating to LSF, e.g. Cernak's Chem. Soc. Rev., 2016, 45, 546-576, although there are several others.

Response: We changed it and added this key reference.

Also, the phrase “[the] reaction was viable with late-stage modification” doesn’t really mean anything. Presumably, the late-stage functionalization was viable.

Response: We corrected it.

Line 110: “products can be scalable to generate” is muddled. The reaction is scalable, not the products.

Response: We corrected it.

Line 116: After “demonstrated” please refer the reader to the appropriate scheme.

Response: We added it.

Line 141: N-PtBu should presumably have two tert-butyl groups.

Response: We corrected it.

Line 162: Should “Association” be “Coordination”?

Response: Yes. We corrected it.

Line 170: “the Si-Si bond of 2a”. In Figure 4a, 2a is shown and has no Si-Si bond.

Response: We corrected it.

Line 184: “compatible” should be “compatibility” and “a broad range of substrate scope” is redundant; “broad substrate scope” is meant.

Response: We corrected them.

SUPPORTING INFO

There are problems in the rigor with which the SI is prepared. For example, in the section describing the synthesis of complex 23, the complex is labelled as 6, which is not consistent with the manuscript.

Response: We corrected the mistakes in SI.

The description of this procedure for 23 (or “6”) on page S27 says “The single crystals of 6 were grown by slow evaporation of the DCM/ Hexane.” What DCM? None is mentioned.

Response: We corrected it and added the full name of DCM.

For clarity, the spectra in the SI should be labelled. For example, for on page S79, I’m guessing that’s 31P NMR and the spectrum on S80 is 19F NMR.

Response: Yes. We corrected it.

The 1H NMR spectrum of 2u on page S86 shows significant impurities, which presumably puts the yield of 2u (86%) in question. Also, what I presume is the 31P NMR of 2u (p S87) is very weak, and I note that the solvent peaks in the 1H and 13C NMR spectra of that compound are proportionally higher than for some of the other compounds. Similarly, the 1H NMR spectrum of

16 is not clean enough, so the yield for that is also presumably not the 74% reported.

Why is the spectrum at the top of page S98 zoomed into such a narrow ppm range? The whole spectrum should be showed - and the nucleus indicated - and an expansion provided if desired.

Response: According to your suggestions, the mentioned products 2u and 16 have been purified. In addition, other compounds like 2c and 2h were also purified during the revision. The yields of these products were updated as well.

I stress that these are all *examples* of problems in the SI. Obviously, it's not possible to list all the others here. But those I listed above are representative and were quick to find, so the entire SI should be carefully revised and checked for rigor in every detail. If yields are not accurate, they should be recorded correctly.

Response: Thanks for this kind suggestion! We have carefully checked and revised the entire SI.

Reviewer #2 (Remarks to the Author):

This manuscript present the work on phosphorus(III)-assisted regioselective C-H silylation of heteroarenes studied experimentally and computationally, which includes that a cyclopalladated intermediate is first confirmed by X-ray analysis to support P(III)-assisted C-H silylation mechanism. The work seems publishable, since this is one type of important reactions. However, authors should consider the following concerns:

Response: Thanks for such positive comments.

(1) For the DFT calculation, authors should address why zero yield is observed for reactants 7-10 in entry 8 of Table 1. At the same time, it will be much better to address the zero yield in entry 9 of Table 1.

Response: Thanks for your suggestion! Based on the DFT calculations shown in Fig. S1, the P^tBu_2 is the best directing group to activate the C7 position with a stable five member formed. This part has been added in the SI.

Fig. S1. C-H activation step using 7, 8, 9 and 10.

(2) Authors should also complete the DMBQ-involved process, if too much for text, it could be

included in SI, which might help to understand the data of entry 2-4 in Table 1.

Response: Thanks for your suggestion! In the presence of $\text{Pd}(\text{OAc})_2$, DMBQ could act as a suitable oxidant to regenerate the $\text{Pd}(0)$ species, and the use of other oxidants like Ag_2CO_3 or $\text{Cu}(\text{OAc})_2$ showed much less efficient for C7-silylation, along with large amount of the by-product 5. To further understand the DMBQ-involved process, we added the experiments and found DMBQ could be an external base to assist the deprotonation step without acetate in the system. Using the DMBQ as the base in the presence of PdCl_2 for C-H activation is feasible with a lower free energy barrier of 26.1 kcal/mol. However, compared to the CMD process, the external base way is impossible using $\text{Pd}(\text{OAc})_2$ as the catalyst with a high free energy of 38.8 kcal/mol. This part has been added in the supporting information.

(3) Authors should compare B3LYP results with M06.

Response: Thanks for this kind suggestion! The DFT calculations are done under the level of M06/6-311++G(d,p), SDD for Pd, SMD(toluene)// B3LYP-D3/6-31G(d), lan12dz for Pd, with Grimme correction for entropy and Head-Gordon correction for enthalpy in 120°C, which is consistent well with the experimental results. When use B3LYP, the conclusion is similar with the result of m06.

Table 1. The relative free energies under the level of B3LYP-D3/6-31G(d), lan12dz for Pd, with Grimme correction for entropy and Head-Gordon correction for enthalpy in 120°C.

	path A	path B
$1/3[\text{Pd}(\text{OAc})_2]_3$	0.0	
INT1	9.9	
TSI	27.5	33.0
INT2	3.2	14.2
TSII	33.5	38.4

INT3	19.4	19.4
TSII	31.4	
INT4	2.1	
TSIV	12.6	
INT5	0.7	
3aa	-26.9	

REVIEWERS' COMMENTS

Reviewer #1 (Remarks to the Author):

I recommend publication after the correction of the errors referred to in the attached .pdf file.

Reviewer #2 (Remarks to the Author):

Authors have replied my concerns, so my suggestion is that it is publishable in Nature Comm.

Reviewer #1 (Remarks to the Author)

This manuscript has been improved since the previous version, especially in the technical details, nomenclature and issues with the SI. I am grateful to the authors for improving the manuscript according to my previous suggestions and for clarifying several of my questions.

Minor corrections remain to be made, especially in two aspects:

1. What the authors refer to as “late-stage functionalization” is not late-stage functionalization and it would be inaccurate to publish it with that label. It is only the functionalization of larger substrates. (Otherwise, the substrates leading to **12** are at the late stage of what?) To me they look like indoles with bulky groups attached by the authors. As far as the authors describe, these are not APIs, advanced intermediates, drug candidates or any other type of functional molecule. They are contrived, bulky substrates only and – as I wrote in the previous round of comments – the steroid units do not demonstrate any additional functional group tolerance compared to the rest of the scope.

I refer the authors to Börgel and Ritter’s recent Perspective on this, where they discuss when the use of the term “late stage” in the literature is appropriate.

Chem. **2020**: <https://doi.org/10.1016/j.chempr.2020.07.007>

2. The article is still full of ambiguous phrases and grammatical errors, even though the authors say it has been proof-read by a native English speaker. The minor grammatical errors (e.g. missing definite articles) are not so important, as I expect them to be corrected during the editing stage. However, some phrases are still confusing to the narrative of the paper and in some cases make it unclear what the authors mean.

I would recommend publications once corrections according to points 1 and 2 are made, as well as the minor scientific points below.

Scientific points:

- Table 1 shows “variation from standard conditions”. There is no set of conditions labelled or described as “standard”. This would help. Presumably, the authors mean the conditions under the reaction arrow at the top of Table 1. But this is not stated. There, the equivalents of DMBQ are mentioned, but not for Si Me₃, which is inconsistent.

Response: The equivalents of Si Me₆ are mentioned under the reaction arrow. In addition, the standard conditions are stated in the text.

What I mean about the standard conditions is that if you have a table column labelled “Variation from standard conditions”, it would be useful to the reader to indicate which conditions the authors consider “standard”. Please label the conditions above the reaction arrow of Table 1, “standard conditions” for the reader’s reference and for clarity.

Lines 87-89: “As expected, rigorous control experiments demonstrate that the reaction parameters including external oxidant DMBQ (entry 11) and palladium catalyst (entry 12) were crucial for the silylation to occur.” This is obviously inaccurate, since the authors show that the products could be obtained without DMBQ (e.g. entry 2 or 3), so it is not “crucial”. It’s just optimal.

Response: We changed the description in the text.

In the version of the revised manuscript I received, it still says “crucial” (line 91). It’s even highlighted.

- Were C2 or C3 silylated products ever observed with the other directing groups? If the more weakly coordinating DGs were not good at directing the Pd to C7, maybe C3-palladation occurred? Maybe it occurred and no silylation occurred? Were any experiments performed to test this?

Response: We didn’t observe any C-H silylation products when using indoles bearing *N*-P(O) Bu₂ (**7**), *N*-Piv (**8**) and *N*-SMe (**10**) groups. While indoles bearing *N*-pyrimidin-2-yl (**9**) was selected as a substrate, a small amount of C2 silylation by-product was obtained.

Please mention that **9** gave small amounts of C2-silylation in the text. To my knowledge substrates like **9** have not previously been successful in C-H silylations, so this is an interesting result.

- In Figure 4C, the KIE is expressed using uppercase “K” letters (which are used to indicate equilibria normally) but refers to rates which therefore should be represented with italic lowercase “*k*” letters. Any other instances of this should be corrected.
- Acetates coordinated to Pd in Fig 5 are written as OAC, rather than OAc.

Examples of unclear phrasing to correct:

Example 1.

Lines 20-21: “A cyclopalladated intermediate was first confirmed by X-ray analysis.”

What does “first” mean? This is ambiguous. Was this the first thing done? Is it the first such complex? Moreover, the sentence could be taken to mean that the intermediacy of the complex was confirmed by X-ray analysis, which presumably the authors don’t mean.

Response: We changed the description.

The new formulation is equally ambiguous:

Line 22:

“the novel cyclopalladated intermediate was successfully synthesized and confirmed by X-ray analysis in order to support the P(III)-directed C–H metalation event.”

The structure of the complex was confirmed, not the complex.

Also, the confirmation of the structure doesn't support the metalation event, it supports the hypothesis about the metalation event. Finally, the analysis wasn't performed in order to support the hypothesis, but to *test* the hypothesis.

Example 2:

“In addition, the substrate scope could be extended to other electron-withdrawing groups like acetyl (**1q**), ester (**1r** and **1s**) and the cyano group having indoles (**1t-v**) was successfully furnished **2q-v** in good to excellent yields.”

The highlighted part of the quote is grammatically challenged. Does “cyano group having indoles” refer to indoles substituted with cyano groups? Also the plural (indoles) is treated as a singular (was, not were) and the position of the “**2q-v**” in the sentence seems wrong. This is another difficult-to-follow sentence about the technical side of the chemistry.

Example 3

Line 121+:

“For instance, C7 Si-directed cross-coupling of ...”

The coupling of **4** at the C-Si carbon is *not* Si-directed. In the same way that Suzuki couplings are not B-directed. The whole manuscript should be carefully proof-read for all of these inaccuracies.

Example 4

Line 124+

“The effective method to form boronate ester **17** by Si-B exchange was also demonstrated in good yield”

This sounds like the demonstration was high yielding, when in fact it was the reaction. It should be “...was demonstrated to proceed in good yield.”

Example 5

Line 11-12:

“an important role in construction of downstream compounds used in natural products...”

What does it mean to say that downstream compounds are used in natural products? Presumably, the authors mean natural product synthesis. Also, it should be “**the** construction”.

Examples of minor language errors

Here are examples of minor language errors. There are many of them, but I include these just for convenience of finding the ones I saw already – *I did not list them all*.

From the end of the paper:

“Because of the ubiquity of the indole and carbazole frameworks and their precursors in biologically active compounds, we hope that this strategy based on the diversity of silicon chemistry will simplify the synthesis and structural elaboration of heteroarylsilanes for advance

research in chemistry, biology and medicine.”

This is an unnecessarily long sentence, but it’s also unclear what “for advance research” is intended to mean. Does it mean:

1. “for advance research” or perhaps
2. “**to** advance research”?

Line 14:

“occurs” should be occur.

Line 17:

Should be “the palladium catalyst”

Line 20:

“The obtained silylated indoles have readily employed”

Should be “have **been** readily employed”

Also: what does it mean for something to be “readily employed” rather than just employed?

Line 23:

“A series of mechanistic experiments and density functional theory (M06-2X) calculations have shown the”

The “have” should be “has”. A series is one thing, so a series of experiments *has* shown, not have shown.

Line 27:

“Silylation of C-H bond” should be “**the** C-H bond”

And (same line): “**the** C–H functionalization field”

Line 55: “**an** efficient manner”

“Analogous to the observation by Sanford,⁶⁷⁻⁶⁹ we proposed”

Should be “By analogy to Sanford’s observation...”

“reductiveelimination” should not be one word.

Reviewer #1 (Remarks to the Author)

This manuscript has been improved since the previous version, especially in the technical details, nomenclature and issues with the SI. I am grateful to the authors for improving the manuscript according to my previous suggestions and for clarifying several of my questions.

Minor corrections remain to be made, especially in two aspects:

1. What the authors refer to as “late-stage functionalization” is not late-stage functionalization and it would be inaccurate to publish it with that label. It is only the functionalization of larger substrates. (Otherwise, the substrates leading to 12 are at the late stage of what?) To me they look like indoles with bulky groups attached by the authors. As far as the authors describe, these are not APIs, advanced intermediates, drug candidates or any other type of functional molecule. They are contrived, bulky substrates only and – as I wrote in the previous round of comments – the steroid units do not demonstrate any additional functional group tolerance compared to the rest of the scope.

I refer the authors to Börgel and Ritter’s recent Perspective on this, where they discuss when the use of the term “late stage” in the literature is appropriate.

Chem. 2020: <https://doi.org/10.1016/j.chempr.2020.07.007>

Response: We removed the term “late-stage functionalization” from manuscript according to your suggestion.

2. The article is still full of ambiguous phrases and grammatical errors, even though the authors say it has been proof-read by a native English speaker. The minor grammatical errors (e.g. missing definite articles) are not so important, as I expect them to be corrected during the editing stage. However, some phrases are still confusing to the narrative of the paper and in some cases make it unclear what the authors mean.

Response: We have carefully checked and revised the entire manuscript.

I would recommend publications once corrections according to points 1 and 2 are made, as well as the minor scientific points below.

Scientific points:

- Table 1 shows “variation from standard conditions”. There is no set of conditions labelled or described as “standard”. This would help. Presumably, the authors mean the conditions under the reaction arrow at the top of Table 1. But this is not stated. There, the equivalents of DMBQ are mentioned, but not for Si₂Me₃, which is inconsistent.

What I mean about the standard conditions is that if you have a table column labelled “Variation from standard conditions”, it would be useful to the reader to indicate which conditions the authors consider “standard”. Please label the conditions above the reaction arrow of Table 1, “standard conditions” for the reader’s reference and for clarity.

Response: The “standard conditions” was stated in footnote of Table 1.

Lines 87-89: “As expected, rigorous control experiments demonstrate that the reaction parameters including external oxidant DMBQ (entry 11) and palladium catalyst (entry 12) were crucial for the

silylation to occur.” This is obviously inaccurate, since the authors show that the products could be obtained without DMBQ (e.g. entry 2 or 3), so it is not “crucial”. It’s just optimal.

In the version of the revised manuscript I received, it still says “crucial” (line 91). It’s even highlighted.

Response: We changed it.

- Were C2 or C3 silylated products ever observed with the other directing groups? If the more weakly coordinating DGs were not good at directing the Pd to C7, maybe C3-palladation occurred? Maybe it occurred and no silylation occurred? Were any experiments performed to test this?

Response: We didn’t observe any C-H silylation products when using indoles bearing NP(O)tBu₂ (7), N-Piv (8) and N-SMe (10) groups. While indoles bearing N-pyrimidin-2-yl (10) was selected as a substrate, a small amount of C2 silylation by-product was obtained.

Please mention that 10 gave small amounts of C2-silylation in the text. To my knowledge substrates like 10 have not previously been successful in C-H silylations, so this is an interesting result.

Response: We added it according to your suggestion.

- In Figure 4C, the KIE is expressed using uppercase “K” letters (which are used to indicate equilibria normally) but refers to rates which therefore should be represented with italic lowercase “k” letters. Any other instances of this should be corrected.

Response: We corrected it.

- Acetates coordinated to Pd in Fig 5 are written as OAC, rather than OAc.

Response: We corrected it.

Examples of unclear phrasing to correct:

Example 1.

Lines 20-21: “A cyclopalladated intermediate was first confirmed by X-ray analysis.”

What does “first” mean? This is ambiguous. Was this the first thing done? Is it the first such complex? Moreover, the sentence could be taken to mean that the intermediacy of the complex was confirmed by X-ray analysis, which presumably the authors don’t mean.

Response: We changed the description.

The new formulation is equally ambiguous:

Line 22: “the novel cyclopalladated intermediate was successfully synthesized and confirmed by X-ray analysis in order to support the P(III)-directed C–H metalation event.” The structure of the complex was confirmed, not the complex. Also, the confirmation of the structure doesn’t support the metalation event, it supports the hypothesis about the metalation event. Finally, the analysis wasn’t performed in order to support the hypothesis, but to test the hypothesis.

Response: We corrected it.

Example 2:

“In addition, the substrate scope could be extended to other electron-withdrawing groups like

acetyl (1q), ester (1r and 1s) and the cyano group having indoles (1t-v) was successfully furnished 2q-v in good to excellent yields.”

The highlighted part of the quote is grammatically challenged. Does “cyano group having indoles” refer to indoles substituted with cyano groups? Also the plural (indoles) is treated as a singular (was, not were) and the position of the “2q-v” in the sentence seems wrong. This is another difficult-to follow sentence about the technical side of the chemistry.

Response: We corrected it.

Example 3

Line 121+:

“For instance, C7 Si-directed cross-coupling of 4...”

The coupling of 4 at the C-Si carbon is not Si-directed. In the same way that Suzuki couplings are not B-directed. The whole manuscript should be carefully proof-read for all of these inaccuracies.

Response: We corrected it.

Example 4

Line 124+

“The effective method to form boronate ester 17 by Si-B exchange was also demonstrated in good yield” This sounds like the demonstration was high yielding, when in fact it was the reaction. It should be “...was demonstrated to proceed in good yield.”

Response: We corrected it.

Example 5

Line 11-12:

“an important role in construction of downstream compounds used in natural products...”

What does it mean to say that downstream compounds are used in natural products? Presumably, the authors mean natural product synthesis. Also, it should be “the construction”.

Response: We corrected it.

Examples of minor language errors

Here are examples of minor language errors. There are many of them, but I include these just for convenience of finding the ones I saw already – I did not list them all.

Response: Thanks!

From the end of the paper:

“Because of the ubiquity of the indole and carbazole frameworks and their precursors in biologically active compounds, we hope that this strategy based on the diversity of silicon chemistry will simplify the synthesis and structural elaboration of heteroarylsilanes for advance research in chemistry, biology and medicine.”

This is an unnecessarily long sentence, but it's also unclear what “for advance research” is intended to mean. Does it mean:

1. “for advanced research” or perhaps

2. "to advance research"?

Response: We corrected it.

Line 14:

"occurs" should be occur.

Response: We corrected it.

Line 17:

Should be "the palladium catalyst"

Response: We corrected it.

Line 20:

"The obtained silylated indoles have readily employed"

Should be "have been readily employed"

Also: what does it mean for something to be "readily employed" rather than just employed?

Response: We corrected it.

Line 23:

"A series of mechanistic experiments and density functional theory (M06-2X) calculations have shown the" The "have" should be "has". A series is one thing, so a series of experiments has shown, not have shown.

Response: We corrected it.

Line 27:

"Silylation of C-H bond" should be "the C-H bond"

And (same line): "the C-H functionalization field"

Response: We corrected it.

Line 55: "an efficient manner"

"Analogous to the observation by Sanford, 67-69 we proposed" Should be "By analogy to Sanford's observation..."

Response: We corrected it.

"reductive elimination" should not be one word.

Response: We corrected it.